# Multielemental, Nutritional, and Proteomic Characterization of Different *Lupinus* spp. Genotypes: A Source of Nutrients for Dietary Use

**DOI:** 10.3390/molecules27248771

**Published:** 2022-12-10

**Authors:** Alfio Spina, Rosaria Saletti, Simona Fabroni, Antonio Natalello, Vincenzo Cunsolo, Michele Scarangella, Paolo Rapisarda, Michele Canale, Vera Muccilli

**Affiliations:** 1CREA—Consiglio per la Ricerca in Agricoltura e L’analisi dell’Economia Agraria, Centro di Ricerca Cerealicoltura e Colture Industriali, Corso Savoia 190, 95024 Acireale, Italy; 2Laboratory of Organic Mass Spectrometry, Department of Chemical Sciences, University of Catania, Viale A. Doria 6, 95125 Catania, Italy; 3CREA—Consiglio per la Ricerca in Agricoltura e L’analisi dell’Economia Agraria, Centro di Ricerca Olivicoltura, Frutticoltura e Agrumicoltura, Corso Savoia 190, 95024 Acireale, Italy; 4Di3A—Dipartimento di Agricoltura, Alimentazione e Ambiente, University of Catania, Via Valdisavoia, 5, 95123 Catania, Italy; 5ICQ-RF—Ispettorato Centrale Qualità e Repressione Frodi, Laboratorio di Catania, Via Alessandro Volta 19, 95122 Catania, Italy

**Keywords:** lupin species, cultivars, micro- and macronutrients, proteins, supercritical CO_2_ defatting process, fatty acid profile, tocopherol concentration, two-dimensional electrophoresis

## Abstract

Among grain pulses, lupins have recently gained considerable interest for a number of attractive nutritional attributes relating to their high protein and dietary fiber and negligible starch contents. The seeds of *Lupinus albus* (cv. Multitalia and Luxor, and the Modica ecotype); *L. luteus* (cv. Dukat, Mister, and Taper); and *L. angustifolius* (cv. Sonet) analyzed in this study were deposited within the germplasm collection of the Research Centre for Cereal and Industrial Crops of Acireale and were sowed in East Sicily in 2013/14. The collected seeds were analyzed for their multielemental micro- and macronutrient profiles, resulting in a wide variability between genotypes. Lupin seed flour samples were subjected to a defatting process using supercritical CO_2_, with oil yields dependent on the species and genotype. We determined the fatty acid profile and tocopherol content of the lupin oil samples, finding that the total saturated fatty acid quantities of different samples were very close, and the total tocopherol content was about 1500.00 µg/g FW. The proteomic analysis of the defatted lupin seed flours showed substantial equivalence between the cultivars of the same species of *Lupinus albus* and *L. luteus*. Moreover, the *L. angustifolius* proteome map showed the presence of additional spots in comparison to *L. albus,* corresponding to α-conglutins. Lupin, in addition to being a good source of mineral elements, also contributes vitamin E and, thanks to the very high content of gamma-tocopherols, demonstrates powerful antioxidant activity.

## 1. Introduction

During the last decade, functional foods have gained attention from the food industry because of their possible role in the maintenance of health and wellbeing [1]. Grain pulses, protein rich and affordable foods, are grown both for seed production and consumption and contribute to a balanced diet. Among grain pulses, lupins have recently gained considerable interest for a number of attractive nutritional attributes relating to their high protein and dietary fiber and negligible starch contents [2]. Lupin seeds are a food ingredient characterized by good nutritional value and technological flexibility. Their high protein content makes lupin seeds an alternative protein source that can substitute for soy, egg, and other animal proteins, and recent clinical investigations have indicated that the consumption of lupins may be beneficial due to their hypocholesterolemic [3,4,5], anti-atherogenic [6], hypotensive [7], and hypoglycemic activities [8,9].

The genus *Lupinus* comprises more than 170 species that have been described in the Old World and only 12 species from Europe [10]. Domesticated lupins include only four species: *L. albus* L. (white lupin), originating from the Mediterranean costal region and grown in the Mediterranean area and Southern Europe; *L. angustifolius* L. (narrow-leafed lupin, blue lupin), widely grown in South-Western Australia; *L. luteus* L. (yellow lupin) grown in Central Europe; and *L. mutabilis* (Tarwe, Andean, or Pearl lupin), originating from South America and grown in the Andes (Chile and Peru). White lupin is currently the only species grown for human nutrition, but narrow-leafed lupin is increasingly used for animal feed, as well as human food.

Certain elements are essential for plant growth and animal and human health. However, if present in excessive concentrations, they become toxic. Zelalem and Chandravanshi [11] determined the concentration of macro- (K, Na, Mg, Ca); micro- (Cr, Mn, Fe, Co, Ni, Cu, Zn); and toxic (Pb, Cd) elements in dry raw and processed white lupin samples grown in Ethiopia. The authors found that the concentrations of all analyzed elements were higher in raw compared to processed white lupin samples, except for Ca, Co, Cu, and Zn, which appeared at higher concentrations in the processed samples than the raw white lupin samples from all of the sampling sites. The authors concluded that raw (dried) and processed *Lupinus albus* could be good sources of major, minor, and trace elements essential to humans. Other authors [12] studied yellow and blue lupin genotypes and found the highest contents of Mn, Fe, Zn, and Se in *Lupinus luteus* seeds. The authors concluded that the concentration of nonessential elements in the tested lupin seeds was far below the levels that would cause a health risk.

Lupin seeds are commonly used for dietary consumption due to their valuable traits, such as their high protein content, beneficial relative protein profile, high dietary fiber content, and favorable fat content. Although most of the studies available in the literature have focused on the seeds’ protein quality, attention has also been paid to their oil content and quality. Several studies have shown that the amount of oil in lupin seeds varies depending on the genotype. Andean lupin showed a considerable amount of oil in its seeds, ranging from 16 to 20% [13], while lower amounts (9.76–12.1%) were recorded for white lupin. The oil contents in narrow-leafed and yellow lupin [12] were found to be less representative, at 6.5 and 5.1%, respectively [14]. The quality rather than the quantity of oil contained in lupin seeds is much more relevant in terms of dietary purposes and applications for both humans and animals. In general, the fatty acid profile and the ratios between individual acids are used as markers of the overall seed oil quality. It has previously been demonstrated that lupin seed oil has a balanced fatty acid composition, with an average total unsaturated fatty acid percentage of 90% [15] and interesting phytosterol, triterpene alcohol, and phospholipid contents [16]. It must be pointed out that the quali-quantitative composition of the seed oil can vary greatly depending on factors such as the genotype, ripening phase, growing environment, pedoclimatic conditions, cultural practices, and extraction technology and/or solvents used. In the past decade, several studies have been performed to determine the effects of genotype and environment on lupin seed oil quality [16,17,18,19,20,21,22]; however, to the best of our knowledge, no data are available on the effects of the extraction technology on the oil yield. Supercritical carbon dioxide (SC-CO_2_) has been employed to extract seed oils with encouraging results for several decades [23,24,25], and it is widely recognized as a green and environmentally friendly alternative to organic solvent extraction for the production of vegetable fats and oils. Taking into consideration the fact that lupin seeds are recognized as a protein-rich crop and their use as a dietary protein source is of increasing interest, the employment of SC-CO_2_ extraction as a defatting process could address consumers’ demands for natural proteins isolated from natural resources through chemical-free processes.

Lupin seed proteins are classified as albumins (water-soluble), globulins (salt-soluble), prolamins (alcohol-soluble), and glutelins (acid/alkali-soluble). Globulins (frequently referred to as conglutins, including in the remainder of this paper) constitute the major fraction of lupin seed proteins, followed by albumins (the globulin-to-albumin ratio is 9:1), while prolamins and glutelins contribute a negligible amount. Based on the sedimentation coefficient, lupin seed storage proteins are categorized as 2S (generally termed albumins), 7S, and 11S (sub-classes of conglutins). Conglutins are classified into four families: α-, β-, γ-, and δ-conglutins. α-conglutins, members of the “legumin-like” or 11S globulin family, comprise hexamers of high molecular weight (acidic subunits), belonging to the N-terminal, and low molecular weight (basic subunits), related to the C-terminal region. These are linked by disulphide bonds [26], with the high molecular weight subunits being glycosylated [27]. Β-conglutins, also referred to as “vicilin-like” or 7S globulins, are trimeric proteins of 16–70 kDa monomers, characterized by the absence of any disulphide linkage and a great extent of glycosylation. Both α- and β-conglutins originate from proteolytic cleavages of precursor molecules. Γ-conglutins, accounting for about 5% of the total protein content, may not fall into the classical seed storage category, as these proteins are not cleaved during seed germination [28]. Lupin γ-conglutin, a 7S protein, is a homo-tetrameric basic glycoprotein with a relative molecular mass of around 47 kDa, whose monomeric units consist of two disulphides linked to heterogenous subunits of 17 and 29 kDa [29]. γ-conglutin displays unique properties: since its amino acid sequence does not match any other legume protein canonical sequence [30], it binds divalent metal ions, especially Zn^2+^ and Ni^2+^ [31], and carries one N-linked oligosaccharide chain [32]. The last group of the lupin seed conglutin family, δ-conglutins, has been the least studied: these are small proteins, similar to other 2S sulphur-rich albumins, comprising two chains of 4 and 9 kDa linked by disulphide bridges. Recently, the complexity of these four conglutin families has been investigated in different lupin species in terms of their gene structure, phylogenetic relationship, and relative RNA and protein abundance during seed development: the results of this investigation demonstrated differential conglutin RNA expression across lupin species with a high level of γ-conglutins in *Lupinus albus* and *L. cosentinii* [33]. γ-conglutins play an important role due to their health-promoting benefits. It has been demonstrated that purified or enriched γ-conglutin lowered blood glucose in hyperglycemic rats [34] and had a substantial hypoglycemic effect in a glucose-overload trial in healthy humans and rats [35]. γ-conglutin is therefore a potential antidiabetic agent [34], and a patent has been deposited for its use in the treatment of type II diabetes [36]. The molecular characterization of bioactive food components is necessary for understanding the mechanisms of their beneficial or detrimental effects on human health, and mass spectrometry is the leading technique for tracing and quantifying the differential expression of specific proteins such as bioactive or allergenic proteins. The knowledge of the protein composition is helpful for understanding the relationship between the protein content and the nutritional and technological properties of a foodstuff [37], and mass spectrometry coupled to electrophoresis and chromatographic separation is the method of choice for protein food analysis [38,39,40].

The present study comprised a comparative multielemental, nutritional, and proteomic characterization of different genotypes of *Lupinus* ssp. (*L. albus*, *L. angustifolius*, and *L. luteus*) grown in the same environmental conditions, with respect to a commercial lupin flour (*L. albus*, cv. Multitalia) available on the food and feed market. The considerable diversity of the Lupinus genus is still widely unexplored and unexploited regarding its potential for improving health and wellbeing. Useful data for the optimization of the SC-CO_2_ extraction process to obtain lupin seed oil and protein isolates of good quality are reported. The fatty acid and tocopherol profiles of the obtained oils are also presented herein in order to promote their use as an ingredient for human and animal diets. Moreover, physiological and health-promoting effects are highly influenced by protein structure. Therefore, an analysis of the diversity of lupin seed proteins was conducted to provide a deeper understanding of the relationships between structure and bioactive properties in order to improve lupin seeds’ nutraceutical potential. For this reason, a proteomic approach involving 2D-PAGE separations and mass spectrometric analysis was performed to obtain the protein profiles of the different genotypes and to evaluate the presence of protein groups characteristic of specific genotypes that might be employed as markers in future investigations.

## 2. Results

### 2.1. Micro- (Cu, Fe, Mn, Zn) and Macronutrient (K, P, Ca, Mg, Na) Profiles

Table 1 reports the micronutrient levels in different genotypes of *Lupinus* spp. (*L. albus*, *L. angustifolius*, *L. luteus*), expressed in mg/kg.

The content of Cu was equal to 85 mg/kg and 70 mg/kg in cv. Multitalia (*L. albus*) and Taper (*L. luteus*), respectively. The observed values were significantly higher for the remaining varieties, which showed close average values between 6.5 and 17 mg/kg. The highest Fe concentrations were found in cv. Dukat (*L. luteus*—200 mg/kg) and cv. Taper (*L. luteus*—160 mg/kg), while the Mn content was higher in cv. Multitalia (*L. albus*—235 mg/kg) and cv. Luxor (*L. albus*—185 mg/kg). The highest Zn concentration was found in the Taper (*L. luteus*) cultivar (90 mg/kg). Fe and Mn showed significant variability in their concentrations (40–200 mg/kg and 30–235 mg/kg, respectively), probably due to differences between species, cultivars, and field conditions.

Table 2 reports the analyses of the macronutrients (K, P, Ca, Mg, Na) in different genotypes of *Lupinus* spp. (*L. albus*, *L. luteus*, *L. angustifolius*), expressed in mg/kg.

The potassium content ranged from 690 mg/kg in cv. Dukat (*L. luteus*) to 9015 mg/kg in cv. Taper. The phosphorus content ranged from 5385 mg/kg in cv. Multitalia to 6825 mg/kg in cv. Taper. The magnesium content ranged from 1815 in cv. Dukat to 2825 mg/kg in cv. Mister (*L. luteus*). The calcium content had a minimum value of 1620 mg/kg in the Modica ecotype (*L. albus*) and a maximum value of 3200 mg/kg in cv. Taper. The sodium contents were very similar, except for the lowest value, which was found in cv. Mister (310 mg/kg), and the highest value, which was found in cv. Taper (945 mg/kg). These data show that the Taper cultivar (*L. luteus*) contained the highest average amount of nutrients, with the potassium values being particularly high compared to those of the other cultivars. For the remaining genotypes, the contents of phosphorus, magnesium, calcium, and sodium (with the exception of the Multitalia cultivar) showed very similar average values, unlike the iron, potassium, manganese, and zinc contents, which varied widely. These findings agreed with those reported by Tizazu and Emire [41] for white lupin with regard to the phosphorus, iron, zinc, and calcium content. Similar results were also reported by Karnpanit et al. [42], who evaluated 10 cultivars of blue lupin, finding high variation in calcium, iron, and zinc content between whole seed and dehulled seed.

### 2.2. Protein Content

Figure 1 shows that the protein content was quite uniform between the different samples. The highest value was found in the Modica ecotype (40.8%) and the lowest in cv. Mister (32.75%).

Similar results were reported by Tomczak et al. [43], who evaluated 18 cultivars of blue lupin, finding that the protein content was in the range of 28–41%. Our findings also agreed with those reported by Mierlita et al. for white lupin [44].

### 2.3. SC-CO_2_ Defatting Process

Table 3 shows the experimental results for the lupin seed flour samples of the different genotypes subjected to the SC-CO_2_ defatting process. The oil yield, measured as the amount of extracted oil per 100 g of original sample, differed according to the species and genotype. The highest oil yield was recorded for Multitalia (11.42 ± 0.31 g/100 g DW), while the lowest was observed for cv. Taper (4.78 ± 0.40 g/100 g DW). In general, our data showed that the SC-CO_2_ defatting process produced a higher oil yield in the white lupin genotypes compared to the yellow and narrow-leafed genotypes, with the only exception being the Modica ecotype, whose oil yield was not significantly different from the other yellow and narrow-leafed genotypes. The percentage oil recovery, calculated based on the total fat content of the raw lupin flour samples, ranged between 72.04 ± 1.10 and 60.70 ± 0.55 for the Luxor and Taper genotypes, respectively. The results showed that, on average, the SC-CO_2_ defatting process prevented the total oil recovery from sinking below 60%, with significantly higher percentages for the two white lupin genotypes, namely Luxor and the Modica ecotype, compared to Multitalia and the other yellow and narrow-leafed genotypes. The use of SC-CO_2_ to defat lupin seed flour has certain advantages in the context of using defatted lupin protein isolates and lupin seed oil as food or feed ingredients. Indeed, this process allowed us to obtain lupin oil and lupin protein isolates that were free from residual chemical solvents, thus avoiding the pollution that comes with the large-scale use of chemical and toxic solvents.

### 2.4. Fatty Acid and Tocopherol Profiles

The total saturated fatty acid quantities (Table 4) were very close between the different samples. In terms of composition, the most abundant saturated fatty acid was palmitic acid (C16:0), with higher values in Sonet (11.05 g/100 g) and Mister (9.54 g/100 g) and lower values in Dukat (4.69 g/100 g), followed by the saturated C18 (stearic acid) and C22 (beric acid) fatty acids. Musco et al. [45], in their study on three species of lupin, including several of the varieties considered in our study, also measured the saturated fatty acid values, finding the same variability between the cultivars. In the study conducted by Chiofalo et al. [46], very similar values were found for palmitic acid, except in Mister and Taper, where lower quantities were observed.

The predominant monounsaturated fatty acid (Table 5) was oleic acid (C18:1n9), with higher values in Luxor (49.41 g/100 g) and lower values in Dukat (26.47 g/100 g). In their study, Musco et al. [45] obtained similar values to those presented herein, showing a very high proportion of oleic acid compared to other monounsaturated fatty acids. In Chiofalo et al. [46], the amount of oleic acid obtained was comparable to that measured in our study, except in the case of cv. Taper, for which the authors reported lower values.

Among the main polyunsaturated fatty acids (Table 6), the highest values were those of linoleic acid and alpha linoleic acid (omega 3). The cultivars Mister (39.99 g/100 g), Dukat (38.94 g/100 g), and Sonet (38.18 g/100 g) had the highest values of linoleic acid (C18:2n9,12), while the Modica ecotype (9.95 g/100 g), Taper (9.26 g/100 g), and Luxor (9.10 g/100 g) had the highest alpha-linoleic acid yields (C18:3n9,12,15). In Musco et al. [45], the cultivars Taper, Dukat, and Sonet showed higher values of linoleic acid than those observed in our study; the authors determined that cv. Taper had the highest linoleic acid content, in contrast to our results. Furthermore, Chiofalo et al. [46] also found that Taper had the highest content of linoleic acid compared to the other cultivars, and, in general, their values were slightly higher than those measured in the current study, being very close the alpha-linoleic acid values.

The lipid fraction of the analyzed lupin seed oil samples was rich in essential fatty acids, linoleic and alpha-linolenic, respectively. This makes these species very interesting from a nutritional point of view. The amount of tocopherols in pulses, specifically in lupin, plays an important role in terms of health, as these nutrients are involved in the prevention of cardiovascular and eye diseases. The tocopherol contents measured in the different genotypes of *Lupinus* spp. (*L. albus*, *L. angustifolius*, and *L. luteus*) under investigation are shown in Table 7.

The total tocopherol content, obtained as the sum of the alpha-, gamma-, and delta-tocopherols, showed an average value of about 1300.00 µg/g FW, with maximum values in both Luxor (1930.00 µg/g FW) and the Modica ecotype (1865.00 µg/g FW) and the lowest value in Multitalia (987.00 µg/g FW). Regarding the individual components of the total tocopherol content, the most important details are as follows:-Alpha-tocopherols: max 96.80 µg/g FW in cv. Taper and min 6.80 µg/g FW in cv. Dukat;-Gamma-tocopherols: max 1706.00 µg/g FW in cv. Luxor and min 874.00 µg/g FW in cv. Multitalia;-Delta-tocopherols: max 111.00 µg/g FW in Modica ecotype and min 10.50 µg/g FW in cv. Dukat.

For comparison, we referred to the work of Lampart-Szczapa et al. [47], in which the tocopherol contents of white lupin were studied. Similar results were also reported by Boschin et al. [48], who evaluated six cultivars of white lupin, seven of narrow-leafed lupin, and one of tarwe, finding high variability in alpha- and delta-tocopherols, mainly for white lupin, and moderate variability in gamma-tocopherols for all lupin species.

### 2.5. Proteomic Analysis

Total protein extracts from mature, dry lupin seeds from *Lupinus albus* (cvs Luxor, Modica, and Multitalia); *L. luteus* (cvs Dukat, Mister, and Taper); and *L. angustifolius* (cv. Sonet) were obtained under denaturing and reducing conditions. The extracts were employed to generate 2D-PAGE maps. For each extract, three technical replicates were obtained (Appendix A). The maps displayed in Figure 2 are representative of each extract. At a glance, the similarities among the cultivars belonging to the same species are extremely high, suggesting the absence of qualitative variations between cultivars. Moreover, the maps depict the intrinsic complexity of the protein pattern, with several spots migrating at the same molecular weight and pI, suggesting extensive heterogeneity, as previously reported by Wait et al. [49].

#### 2.5.1. Analysis of *Lupinus albus* Proteome cv. Luxor

To characterize the proteome of the *Lupinus albus* cultivar Luxor, 128 spots of the 2D gel map were excised and digested with trypsin. The peptide mixture from each protein spot was analyzed by HPLC-nESI MSMS. The raw file obtained by each analysis was subjected to a bioinformatic search against the *Viridiplantae* database using the MOWSE algorithm implemented in the Mascot search engine. The details of the parameters employed to identified the proteins and validate the results are reported in the Materials and Methods section. The excised spots are labeled in Figure 3, and the details of protein identification are reported in Appendix A. For some adjacent and faint spots, a single excision was not always possible. Therefore, these spots were excised together and labeled with the two numbers originating from the 2D gel analysis software; in the following discussion, they are considered as a single spot. With only a few exceptions, most of the proteins were identified as α-, β-, γ-, or δ-conglutins. Within each conglutin group, most of the proteins were identified with the same UniProt accession number. Consequently, these spots could be attributed to isoforms of the same protein with differing degrees of post-translational modification. In addition, the identification of proteins with a molecular mass lower than the theoretical value was probably related to the proteolytic cleavages occurring in the precursor molecules. A summary of all the identified conglutins is reported in Table 8. The identified proteins were grouped according to their UniProt accession number and according to the position of the identified peptides in the primary sequence. This analysis was performed by a comparison of the sequences identified by the bioinformatic search and the full primary sequence downloaded.

As far as the β-globulin/vicilin family is concerned, the identified proteins were related to the following three deposited sequence entries: Q53HY0, Q6EBC1, and F5B8W0. The sequence alignment of these three entry proteins showed an extensive sequence identity, ranging between 87 and 94% (Appendix A), thus confirming the extensive sequence similarity of these proteins. Spots were grouped according to the position of the identified peptides in relation to the primary structure of the protein: N-terminal and central region, central region, or central and C-terminal portion (Table 8, Figure 3). Five spots were identified with peptides belonging to the N-terminal and central region: three of them (identified by acc. no. Q53HY0) migrated at around 40 kDa and almost a neutral pI, while the other two (acc. no. Q6EBC1) showed lower molecular masses (30 kDa and 16 kDa) and a basic pI. Most of the β-globulin/vicilin-like proteins were identified with peptides in the central region of the protein sequence: twenty-eight belonging to the Q53HY0 entry and eighteen to the Q6EBC1 entry, with the two sequences differing only in terms of a few amino acid substitutions. Fourteen β-globulin/vicilin proteins were identified with peptides belonging to the central and C-terminal region. Only ten spots did not fall into these categories, due to the random localization of the identified peptides (i.e., spot IDs 302, 183/194, 214, 141/161, 178/173, 309, 311, 323, 364, and 376).

Similar findings were obtained for the α-conglutin/legumin family (i.e., α1 e α2 conglutins, legumin-like proteins, and seed storage proteins), as these proteins were identified through peptides belonging to different regions of the entry sequences with the following acc. nos.: Q53I54, Q53I55, F5B8V6, F5B8V7, and F5B8V8. Sequence alignment revealed that the Q53I55 sequence corresponded to the C-terminal fragment of Q53I54 with very few substitution points (Appendix A). Conversely, the other three sequences (i.e., F5B8V6, F5B8V7, and F5B8V8) showed a sequence identity ranging between 47 and 75% (Appendix A).

As previously reported, 10 α-globulin proteins were hexameric proteins consisting of an N-terminal acidic subunit and a C-terminal basic subunit. Spots 233, 235/254, and 256, identified by peptides belonging to the N-terminal region of the Q53I54 entry, were focused in the acidic region of the 2D map; spots 341 and 346, focused at a lower Mr and less acidic pI, were instead identified through peptides of the central region of the same protein entry as a result of the proteolytic cleavages occurring in the precursor molecules.

Almost all γ-conglutin spots were localized in the basic pI region, and at both a high and low Mr. They were all related to the Q9FSH9 database entry, with the exception of spot 378, which was characterized through peptides of the Q9FEX1 entry. Analogously, all the δ-conglutins were identified by peptides retrieved from a single database entry (i.e., Q333K7).

Moreover, multiple identifications were found in several spots, such as the presence of both α- and β-conglutins (e.g., spot IDs 202 and 233); α- and δ-conglutins (spot ID 341); β- and γ-conglutins (spot IDs 257/260, 261, 262, 294/293, 308, 309, 310, 315, 322, 324, 326, 334, 344, 363/370, 374, and 376); β- and δ-conglutins (spot IDs 350, 364, 373/384, 381, and 386), γ- and δ-conglutins (spot ID 388); and β-, γ-, and δ-conglutins (spot IDs 296 and 389).

In addition, certain proteins other than conglutins, such as proteins belonging to the family of HSPs (spot IDs 357 and 362), quinone oxidoreductase (spot ID 229), and glutathione transferase (spot ID 331), were identified. However, these proteins were identified as gene products of species other than lupin, because the corresponding entries of *Lupinus* spp. were lacking in the database (see Appendix A).

Each group of identified proteins is reported in Figure 2 using the following depictions: solid-line rectangles indicate the acidic and basic subunits of α-conglutin/legumin; dash-and-dot rectangles indicate the β-conglutin/vicilin family protein spots; dotted ovals indicate δ-conglutin/vicilin; solid-line ovals enclose both the large and small subunits of γ-conglutins; and, finally, the solid-line hexagons enclose the subunits of both β- and γ-conglutins.

Taking into consideration all the identifications obtained for the conglutin proteins, the identification of different spots with the same few gene sequences suggested an extensive homology between the different protein families. The presence of different isoforms of the same protein may have arisen from the different degrees of post-translational modification, but their characterization was beyond the aim of this work.

#### 2.5.2. Comparison of *Lupinus albus* cv. Luxor, cv. Multitalia, and Modica Ecotype Proteomes

The comparison of *L. albus* cv. Luxor, cv. Multitalia, and the Modica ecotype was carried out with Luxor as a reference, and image analysis was performed to detect the presence of protein spots with different abundances. The comparison of the 2D maps of the Luxor and Multitalia cv. did not show any spots with differences in relative abundance nor any unique spots. However, the comparison between Luxor cv. and the Modica ecotype revealed some differences (Appendix A). In particular, four spots showed differences in their relative abundance: in the Modica ecotype, three spots (i.e., spot IDs 31*, 41*, and 43*) decreased, whereas spot ID 49* increased. Moreover, four spots (i.e., spot IDs 72*, 75*, 84*, and 112*) appeared to be unique in the Modica ecotype, and two spots (spot IDs 125* and 129*) were exclusively present in the 2D map of the Luxor cv. The corresponding identifications are reported in Appendix A. In more detail, regarding the four unique spots of the Modica ecotype, the MS analysis of the spots 72* and 75* did not allow any identification. However, in spot IDs 84* and 112*, proteins belonging to the conglutin family were identified. In more detail, in spot ID 84 a beta-conglutin (acc. no. Q6EBC1) was identified; in spot ID 112*, a delta-conglutin (Q333K7) and an alpha 2 conglutin (F5B8V7) were detected. Moreover, in spot ID 84*, the presence of a peroxiredoxin was also revealed.

Analogously, the MS analysis of the two unique spots in the Luxor cv. (i.e., spot IDs 125* and 129*) allowed the identification of peptides belonging to the conglutin family together with the same peroxiredoxin already observed in the unique spot ID 84* of the Modica cv. (see Appendix A). In particular, the beta-conglutin with the acc. no. Q6EBC1 was identified in both these spots. In addition, the beta-conglutin/vicilin-like protein (acc. no. Q53HY0) and gamma-conglutin (Q9FSH9) were identified in spots 125* and 129*, respectively. These results allowed us to assume that the beta-conglutin identified in the unique spot ID 84 of the Modica ecotype, migrating at 25 kDa, and the protein identified in the unique spot IDs 125* and 129* of Luxor, focused at different Mr and pI values, probably represented different isoforms of the entry deposited in the database (theoretical Mr 62.1). These isoforms exhibited variations in the unidentified portions of the amino acid sequence, resulting in proteins with different molecular weights and isoelectric points. The same conclusions can be extended to the identified beta-conglutin/vicilin-like protein (acc. no. Q53HY0) and gamma-conglutin (Q9FSH9), whose experimental Mr was lower than that calculated from the deposited sequences.

Overall, from the analysis of the data reported here, it appears that the protein content of the Luxor cultivar, the Modica ecotype, and the Multitalia cultivar did not differ significantly, with substantial equivalence among the cultivars (Figure 2).

For the above reasons, in the subsequent comparative proteomic evaluation of *Lupinus albus* cultivars versus *L. luteus* cultivars, the Luxor cultivar was taken as the reference for *L. albus*.

#### 2.5.3. Analysis of *Lupinus luteus* Proteome

Two-dimensional electrophoretic maps of the three *Lupinus luteus* cultivars (Dukat, Mister, and Taper) are shown in Figure 2. Images analyses were performed in order to obtain a statistical evaluation of the relative abundances of the protein spots and highlight the presence of unique spots. The comparisons were performed using Taper cv. as a reference. The Dukat vs. Taper comparison did not evidence variation in the relative abundance of the spots present in the two 2D maps. However, the presence of six spots exclusive to the Taper cv. (i.e., spot IDs 73*, 74*, 76*, 78*, 79*, and 80*) and two spots uniquely present in the map of the Dukat cv. (spot IDs 88* and 99*) were detected. The spots are labeled in Appendix A, and the results of the protein identification are reported in Appendix A. Among the unique spots of cv. Taper, MS data allowed the characterization of peptides belonging to the central and C-terminal portions of α-conglutin (spot IDs 73* and 78*) and of a peroxiredoxin (spot ID 79*). In addition, in spot ID 74*, peptides belonging to an α-conglutin/seed storage protein, as well as a γ-conglutin, were identified. The analysis of the two spots (spot IDs 88* and 99*) exclusively present in the 2D gel of the Dukat cv. allowed the identification of an annexin, a protein commonly expressed in response to environmental stresses and signaling during the growth and development of plants, as well as a putative seed triacylglycerol (TAG) factor protein co-migrating in spot 99*. No results were obtained for spot ID 88*. In conclusion, regarding the conglutins identified as exclusive to Taper and Dukat, the finding of proteins with lower experimental Mr values than the theoretical value may be attributed to the occurrence of a proteolytic cleavage in the precursor molecules as a result of the maturation process. Thus, based on the protein identification results and 2D gel image analysis, a substantial equivalence between the Dukat and Taper cv. of yellow lupin was determined.

The comparison between the 2D PAGE maps of the cultivars Taper and Mister showed the presence of a unique spot in the Taper cv. (i.e., spot ID 66*) and six unique spots in the Mister cv. (spot IDs 81*, 82*, 83*, 85*, 92*, and 96*). The spots are labeled in Appendix A, and the results of the protein identification are reported in Appendix A. In spot ID 66*, exclusively present in the Taper cv., many peptides related to different annexin entry sequences and two peptides belonging to a conglutin beta protein 5 were identified.

The unique spots of Mister cv. were identified as belonging to the α-conglutin family (spot IDs 81* and 92*), δ-conglutin family (spot IDs 83* and 85*), and a putative TAG factor protein (Spot 96*).

Again, the unique spots corresponded to conglutins that were identified with the same accession number and different Mr values as a result of proteolytic cleavage during the maturation process, and so the analysis of the 2D gel images and the results obtained from the protein identifications allowed us to determine a significant level of equivalence between the Mister and Taper cultivars of yellow lupin.

Consequently, the results presented above revealed no relevant differences among the three yellow lupin cultivars investigated, with the exception of a few unique proteins belonging to the α-, β-, γ-, and δ-conglutin families. However, these unique protein spots did not represent a differentiation marker among the analyzed cultivars. For the above reasons, the Taper cv. was considered as the reference for the yellow lupin cultivars and was thus chosen for the subsequent comparative analysis with the Luxor cv. (which was taken as the reference cultivar for white lupin).

#### 2.5.4. Comparison of *Lupinus albus* vs. *L. luteus* Lupin Proteome

Although a detailed cross-species proteomic comparison was beyond the scope of this work, the image analysis of the 2D gels was performed to compare the *Lupinus albus* (with Luxor cv. as reference) and *L. luteus* (with Taper cv. as reference) proteomes (Figure 2). Taking into account the results obtained by MS, reported in the two previous paragraphs, the image analysis revealed the presence of exclusive spots in *Lupinus albus*. In detail, these differences were related to: (i) a group of β-conglutins (dash-and-dot rectangle) migrating at a basic pI and 30–50 kDa, which were identified with peptides mostly belonging to N-terminal and central region proteins; (ii) a group of spots focused at a basic pI and 18–38 kDa (solid-line hexagon), which predominantly corresponded to β- and γ-conglutins; and finally, (iii) a group of spots migrating at an acidic pI and 12–18 kDa, which corresponded to δ-conglutins (dotted-line oval).

#### 2.5.5. Analysis of *Lupinus angustifolius* Proteome

Two-dimensional electrophoretic maps of the *L. angustifolius* (cv. Sonet) proteome are shown in Figure 2. The image analysis of the *L. angustifolius* proteome map showed the presence of additional spots with respect to *L. albus*. Thus, these protein spots were excised, in-gel digested, and analyzed by mass spectrometry. The excised spots of *L. angustifolius* are labeled in Figure 3b, and the details of the protein identification are reported in Appendix A. As far as the β-globulin/vicilin family is concerned, the identified proteins belonged to different sequence regions of eight proteins (see Appendix A). As for the identifications reported in Table 8, the spots were grouped according to the verified sequence region defined by the identified peptides in the N-terminal and central, central, and central and C-terminal regions (Table 9). Four spots (spot IDs 274, 325, 326, and 332) did not fall into these categories due to the random localization of the identified peptides. Fifteen proteins were identified with peptides belonging to the central region of eight different proteins. The other thirteen β-globulin/vicilin proteins were identified with peptides belonging to the central and C-terminal regions of three β-conglutins. The sequence alignment of the identified proteins showed an extensive sequence identity ranging between 80.2 and 99.8% (Appendix A).

Similar findings were observed for the α-globulin/legumin proteins. The identified peptides almost all belonged to the central and C-terminal regions of the sequence entries F5B8V6, F5B8V7, and F5B8V8, and migrated between a neutral and basic pI. As previously reported in the analysis of *Lupinus albus*, we found multiple identifications in several spots, mostly showing the presence of both β- and α-conglutins (308, 271, 275, 287, and 279/281/297/295). In addition, four protein spots corresponding to γ-conglutins were identified by a single database entry (Q42369) and were localized in the region of 10–15 kDa. The protein spots highlighted with the dash-and-dot rectangles in Figure 2 were not excised, as they were considered to correspond to β-conglutins (see the identification details of *L. albus*).

#### 2.5.6. Comparison of *Lupinus angustifolius* vs. *L. albus* Proteome

An image analysis of the 2D gels was performed to compare the *L. angustifolius* (cv. Sonet) and *L. albus* (cv. Luxor) 2D gel maps (Figure 2). Although a cross-species proteomic comparison was not the aim of this work, the comparison showed: (i) the same group of acidic subunits of α-conglutin; (ii) a higher number of neutral and basic α-conglutins at 27–23 kDa in *L. angustifolius*; (iii) a higher amount of β-conglutins in *L. angustifolius*; (iv) a similar number of spots identified with both β- and γ-conglutins; and (v) the absence in *L. angustifolius* of spots corresponding to δ-conglutins at an acidic pI and 12–18 kDa.

#### 2.5.7. Comparison of *Lupinus angustifolius* vs. *L. luteus* Proteome

The comparative image analysis of the *L. angustifolius* (cv. Sonet) and *L. luteus* (cv. Taper) 2D gel maps (Figure 2) showed: (i) the same group of acidic subunits of α-conglutin (solid-line rectangles at 27–40 kDa); (ii) a higher number of spots, focused at 15–25 kDa, corresponding to both neutral and basic α-conglutin subunits in *L. angustifolius*; (iii) a higher amount of β-conglutins (dash-and-dot rectangles) in *L. luteus*; (iv) the absence in *L. luteus* of spots at a basic pI and 18–38 kDa identified with both β- and γ-conglutins (solid-line hexagons); and (v) the presence γ-conglutins (solid-line ovals) migrated at a neutral pI and lower Mr in *L. angustifolius* compared to *L. luteus*.

## 3. Materials and Methods

### 3.1. Plant Material

Seeds of *Lupinus albus* (cv. Luxor, cv. Multitalia, and Modica ecotype); *L. luteus* (cv. Dukat, Mister, and Taper); and *L. angustifolius* (cv. Sonet) were deposited within the germplasm collection of the Research Centre for Cereal and Industrial Crops of Acireale (Catania, Italy). The field trial was conducted from 2013 to 2014 in East Sicily (Acireale, Catania, Italy) in the ‘S. Salvatore’ Experimental Station of CREA.

All seed samples were sowed in duplicate plots of 5 m^2^ (2.5 m × 2.0 m). Manual seeding was completed on 4 December 2013. Fertilization was applied during sowing with the fertilizer ‘Ortofrutta’, containing 12 kg ha^−1^ of N, 12 kg ha^−1^ of P_2_O_5_, and 12 kg ha^−1^ of K_2_O (Adriatica S.p.A. Loreo, Rovigo, Italy).

Mechanical weed control was applied in post-emergence at the end of January. Aphicide treatment with 50 mL/hL of Imidacloprid (Afidane, Chimiberg-Diachem S.p.A. Caravaggio, Bergamo, Italy) was applied on 3 March 2014.

The crop was harvested on 15 July 2014.

### 3.2. Chemicals

Formic acid (FA), dithiothreitol (DTT), iodoacetamide (IAA), Tris-HCl, urea, glycerol, sodium dodecyl sulfate (SDS), bromophenol blue, phosphoric acid, and ammonium sulfate were obtained from Sigma-Aldrich (Milan, Italy). Colloidal Coomassie Brilliant Blue G-250 was purchased from Amresco, Solon (OH), USA. Modified porcine trypsin was purchased from Promega (Milan, Italy). Water and acetonitrile for LC/MS analyses were provided from Carlo Erba (Milan, Italy).

### 3.3. Milling of Lupin Seeds

Lupin seeds were first broken by an experimental mill, Cyclotec type 120 (Falling Number, Huddinge, Sweden), with a sieve of 0.5 mm, and then the broken seeds were finely ground by a Bimby TM6 Vorwerk Thermomix Robot.

### 3.4. Multielemental Micro- (Fe, Zn, Mn, Cu) and Macronutrient (P, K, Ca, Mg, Na) Profiles

The seeds were milled and homogenized (final particle size ≤ 200 µm). For sample preparation for ICP-OES analysis, 0.5 g of milled lupine seeds to the nearest 1 mg was accurately weighed in a microwave vessel. For trace-element analysis, 2 mL of deionized water, 8 mL of nitric acid (concentration ≥ 69.0%), and 2 mL of extra-pure hydrogen peroxide (30% *w*/*w*) were added, and the samples were left for 12 h for reaction stabilization until the formation of bubbles had finished. The vessel was sealed and heated in the microwave system. A CEM MARS6 instrument was used for microwave digestion. The following thermal conditions were applied: 150 °C was reached in approx. 20 min and maintained for 30 min, and then 200 °C was reached in approx. 20 min and maintained for 30 min for the accomplishment of specific reactions. After cooling, the prepared solution was filtered through a filter with a 0.45 μm pore size. The solution was transferred to a 25 mL volumetric flask and supplemented with water to reach a 25 mL volume. Optima 2100 DV ICP/OES (PerkinElmer) and WinLab32 software for ICP/OES version 3.4.1.0271 (PerkinElmer) were used for analysis. The following operating conditions in Optima 2100 DV ICP/OES were used for the analysis of the samples: spray chamber type—cyclonic; plasma aerosol type—wet; sample flow rate—1.50 mL/min; plasma—15 L/min; aux—0.2 L/min; nebulizer—0.4 L/min; power—1300 Watts.

### 3.5. Protein Content

Nitrogen content was measured using the Kjeldahl method with a factor of 5.7 to determine the protein content. For sample preparation, 1.0 g of milled lupine seeds to the nearest 1 mg was accurately weighed in a sample tube with a size of 300 mL. The digestion of the samples was carried out using 15 mL sulfuric acid (concentration 95–98%) and a tablet of copper (II) sulfate as a catalyst, which oxidized organic matter to carbonic acid. The following thermal conditions were applied: 390 °C was reached in approx. 60 min and maintained for 180 min. A SpeedDigester K-439 (Buchi) instrument was used for the digestion of samples. Noxious fumes generated during digestion were captured by a sealed suction system and neutralized by the optionally available K-415 Scrubber (Buchi). After cooling, the solution in the digestion tube was then made alkaline by the addition of 90 mL of sodium hydroxide (32% *w*/*w*), which converted ammonium sulfate to ammonia gas. Ammonia was released and distilled into an excess (50 mL) of boric acid solution (4% *w*/*w*) adjusted to pH 4.65. The nitrogen content was then estimated by the titration of the ammonium borate formed with a standardized sulfuric acid solution (0.2 N) using a suitable indicator to determine the end point of the reaction. A KjelFlex K-360 distillation unit (Buchi) and KjelFlex K-360 ver 01.03 software were used to distil ammonia.

### 3.6. SC-CO_2_ Defatting Process

The employed supercritical fluid extraction (SFE) equipment is located at the technological pavilion of the CREA Research Center for Olive, Fruit, and Citrus Crops (Acireale, CT, Italy) and is entirely composed of AISI 316 L steel (Separeco s.r.l., Pinerolo, TO, Italy) according to the European Commission regulations for pressure systems (PED—Pressure Equipment Directory). Figure 4 shows a scheme of the plant. It consists of an extraction reactor (E1) and two separators (S1 and S2). The reactor has a volume of 1 L, while the separators have a volume of about 0.2 L each. The CO_2_ pump has a max flow rate of 5.78 L/h. The operating pressures can vary from 60 to 300 bar. The safety valves are therefore calibrated for opening above 330 bar. The system is thermostated using boiler and chiller integrated systems. Full-fat lupin seeds belonging to the different genotypes were previously oven dried at a mild temperature (<45 °C) and further milled to a fine powder by a domestic grinder so that the final particle size was ≤150 µm. Prior to the SC-CO_2_ defatting process, an aliquot of 40 g of dried full-fat lupin seed flour of the different genotypes was used to determine the oil fraction by Soxhlet extraction for 5 h using hexane as the solvent. Then, the obtained lupin seed flour samples were lyophilized before being subjected to the defatting process by SC-CO_2_. In order to improve the oil extraction yield, about 300 g of the lyophilized samples was manually blended with ethanol as an entrainer at a rate of 15% (*w*/*w*), before being placed in E1, previously set at the desired temperature (T = 40 ± 1 °C). CO_2_ (99.95% grade purity) was subcooled in a cooling system in order to be pumped as a liquid by the pump (Lewa, max flow 5.78 L/h) and then directed into E1 until the desired pressure was reached (P = 300 bar). SC-CO_2_ was allowed to flow continuously through E1 for 6 h at a rate of 5.176 kg/h. The oil was collected in a flask immersed in an ice bath at around 5 °C.

### 3.7. Fatty Acid Analysis

Fatty acid profile of lupin oil samples was determined by direct base-catalyzed transesterification and by gas-chromatography (GC). In brief, 50 mg of oil were converted to fatty acid methyl esters (FAME) by base catalyzed transesterification using 1 mL of freshly prepared 0.5 M sodium methoxide in methanol for 3 min at room temperature followed by 5 min of rest. Then, 2 mL of hexane were added and the supernatant was collected and placed in a 2 mL GC vial. The FAME were separated and quantified using a gas chromatograph (TRACE GC, Thermo Finnigan, San Jose, CA, USA) equipped with a flame-ionization detector and a 100-m fused silica capillary column (25-mm i.d., 0.25-μm film thickness; SP-2560; Supelco Inc., Bellefonte, PA, USA) and helium as the carrier gas (1 mL/min). Gas-chromatography conditions and identification of FAME was performed as reported by Natalello et al. [50].

### 3.8. Tocopherol Analysis

Tocopherol concentrations in lupin oil samples were determined as described in Gliszczyńska-Świgło and Sikorska (2004) [38] with some modifications. Briefly, 100 mg of oil was dissolved in 4 mL of 2-propanol and vortexed thoroughly for 1 min. The solution was then filtered using 0.2 µm/13 mm PTFE syringe filters, and an aliquot was placed into a 2 mL amber vial. Tocopherols were separated and quantified through ultra-high-performance liquid chromatograph (UHPLC) as detailed by Natalello et al. [51]. A sample volume of 10 µL was injected into a Nexera (Shimadzu Corporation, Kyoto, Japan) UHPLC system equipped with a C18 phase column (Zorbax ODS; 25 cm × 4.6 mm, 5 µm; Supelco, Bellefonte, PA, USA). The mobile phase was methanol with a flow rate of 1.3 mL/min. The temperature of the samples and the column were adjusted to 25 °C and 40 °C, respectively. Tocopherols were detected using a spectrofluorometric detector at a 295 nm excitation wavelength and a 330 nm emission wavelength. The tocopherols were identified by the comparison of their retention times with those of pure standards (Merck Life Science S.r.l., Milano, Italy). The quantification was achieved by external calibration curves made with pure standards.

### 3.9. Protein Extraction

For two-dimensional (2D) electrophoresis, 50 mg/mL of defatted lupin flour was extracted in 8 M urea, 4% (*w*/*v*) CHAPS, 60 mM DTT, and 2% (*v*/*v*) immobilized 3–10 pH gradient (IPG) buffer (Bio Rad) at room temperature for 3 h. Extracts were centrifuged at 8000× *g* for 30 min at 4 °C, and the supernatant was stored at −20 °C until use. The protein concentration was measured using the Qubit™ Protein Assay kit with the Qubit 1.0 Fluorometer (ThermoFisher Scientific, Milan, Italy) [52].

### 3.10. Two-Dimensional Electrophoresis

IPG strips (pH 3, 10 NL, 11 cm (Ready Strip Bio-Rad)) were passively rehydrated o/n with 200 μg sample protein in the IPG Box (GE Healthcare, Milano, Italy). Isoelectric focusing (IEF) was performed using an Ettan IPGphor 3 Isoelectric Focusing Unit (Ettan IPGphor Manifold; GE Healthcare) for a total of 13.5 kVh, with a maximum of 6 kV, at 20 °C, 65 μA/strip. Focused IPG strips were equilibrated twice in SDS equilibration buffer (50 mM Tris-HCl pH 8.8), 6 M urea, 30% (*v*/*v*) glycerol, 2% (*w*/*v*) SDS, and 0.002% (*w*/*v*) bromophenol blue) containing 65 mM DTT for the first equilibration and 135 mM iodoacetamide for the second. SDS-PAGE was conducted on a Midi-Protean Cell (Bio-Rad) and Any kD™ Mini-Protean^®^ TGX™ Precast gel (13.3 × 8.7 cm) (Bio-Rad, Hercules, CA, USA). Four microliters of protein molecular marker (Page Ruler unstained protein ladder, Thermo Scientific, Waltham, MA, USA) were loaded. Gels were stained using colloidal Coomassie Brilliant Blue G-250 (Amresco) in 10% phosphoric acid and 10% ammonium sulfate. Gels were fixed in ethanol, water, and acetic acid (4:5:1) for half an hour and then stained overnight in colloidal Coomassie. Gels were stored in 7% acetic acid until used for spot excision. A total of 21 2-DE gels, resulting from three technical replicates for each lupin cultivar, were analyzed. The gel images were acquired using an Epson 1680 Pro scanner and uploaded onto the Image Master 2D Platinum ver. 7 (GE Healthcare). Spots in the gels were detected automatically. Only statistically significant protein spots (*p* ≤ 0.01 ANOVA) were taken into consideration when evaluating the differential expression, with a two-fold change between gels considered an increase in relative abundance and a ratio of at least 0.5 considered a decrease in relative abundance. These spots were manually excised and subjected to mass spectrometric analyses. Only spots observed in all three gel sets were considered for protein identification.

### 3.11. In-Gel Digestion and Mass Spectrometric Analysis

Selected protein spots from the 2-DE gels were manually excised and transferred to 1.5 mL microcentrifuge tubes. Low-abundance protein spots were removed from all the replicate gels and pooled. According to the previously reported procedure, the spots were washed and subjected to in-gel trypsin digestion [53,54]. After soaking the gel pieces in trypsin, the supernatant containing excess trypsin was removed, and the gel pieces were covered with 50 μL of 50 mM NH_4_HCO_3_ pH 8.3 and incubated at 37 °C overnight. The enzymatic reaction was stopped by cooling the gel pieces and the supernatant solution at −20 °C. After in-gel digestion, the supernatant solution was transferred into a clean 1.5 mL tube. The peptides were extracted from gel pieces with 50 µL of 5% aqueous FA and subsequently with 50 μL of CH_3_CN. This extraction procedure was repeated three times. The total extracts were pooled, combined with the first supernatant, lyophilized, and redissolved in 15 µL of 5% FA aqueous solution. Capillary RP-HPLC/nESI-MS/MS was performed using an Ultimate 3000 LC system combined with an autosampler and a flow splitter 1:100 (Dionex Corporation, Sunnyvale, CA, USA) coupled online with a linear ion trap nano-electrospray mass spectrometer (LTQ, Thermo Fischer Scientific, San Jose, CA, USA). Ionization was performed with liquid junction using an uncoated capillary probe (30 ± 2 µm i.d.; New Objective, Woburn, MA, USA). Ten microliters of enzymatic digest solution for each spot were loaded onto a C18 μ-precolumn cartridge (0.3 mm × 5 mm, 100 Ǻ, 5 μm, PepMap, Dionex) and equilibrated with 0.5% aqueous FA at a flow rate of 20 μL/min for 4 min. Subsequently, peptides were applied onto a C18 capillary column (0.18 mm × 150 mm, 300 Ǻ, 5 μm, Thermo Electron, Waltham, MA, USA) and eluted at room temperature with a linear gradient of CH_3_CN + 0.5% FA/H_2_O + 0.5% FA from 10 to 50% in 50 min at a flow rate of 1.5 μL/min. Repetitive mass spectra were scanned using the following electrospray ion source parameters: capillary temperature, 220 °C; spray voltage, 1.9 kV. Peptide ions were analyzed by the data-dependent method as follows: (1) full-scan MS in the m/z range 350–2000; (2) zoom scan of the three most intense ions (isolation width: 2 Da); and (3) MS/MS analysis of the three most intense ions (Q 0.250, collision energy 29 a.u.). Mass calibration was conducted using a standard mixture of caffeine (Mr 194.1 Da), MRFA peptide (Mr 523.6 Da), and Ultramark (Mr 1621 Da). Data acquisition was performed using Excalibur v. 1.4 software (Thermo Fisher Scientific).

### 3.12. Database Search and Protein Identification

MS/MS data were used to perform protein identifications by searching in the UniProt “Viridiplantae” database compiled by SwissProt with 25,438 sequences and TrEMBL with 5,724,529 sequences (January 2015 release). The identifications were achieved using the MOWSE algorithm as implemented in the Mascot search engine version 2.5 (Matrix Science: www.matrixscience.com, accessed on 26 January 2015 ). The following parameters were used for database searches: cleavage specificity—trypsin with two missed cleavages allowed; mass tolerance of 1.2 Da for the precursor ions and a tolerance of 0.6 Da for the fragment ions; allowed modifications—Cys carboxyamidomethylation (fixed), oxidation of Met (variable), transformation of N-terminal Gln, and N-terminal Glu residue in the pyroglutamic acid form (variable). In MASCOT searches, protein scores were derived from ion scores as a non-probabilistic basis for ranking protein hits. Only proteins that met the following criteria were accepted as unambiguously identified: (1) at least two identified peptides fully tryptic; (2) MASCOT score > 57 (probability-based MOWSE score: −10*log(P), where P is the probability that an observed match is a random event.; score > 57 indicates identity or extensive homology (*p* ≤ 0.05)). All identifications obtained through MASCOT software were subjected to a manual interpretation of the MS/MS spectra assisted by PepNovo software (http://proteomics.ucsd.edu/Software/PepNovo.html, accessed on 26 January 2015 ) to confirm or disregard peptide sequence assignation and therefore protein identification. Multiple-sequence alignment was performed with ClustalOmega (http://www.ebi.ac.uk/Tools/msa/clustalo/, accessed on 26 January 2015 ).

## 4. Conclusions

The results for the multielemental composition showed wide variability between genotypes. The amounts of mineral elements presented in this study indicate an excellent nutritional profile, suggesting the suitability of lupin as a raw material for consumption. The protein content results were similar for the different species and genotypes.

The employment of innovative green technologies that do not involve the use of organic solvents or other potentially toxic chemicals to defat lupin seed flour is a matter of growing interest, since consumers are increasingly attracted to natural proteins isolated from protein-rich crops. Moreover, lupin seed oil is a product of much relevance due to its fatty acid and tocopherol profiles, which encourage its use as a valuable ingredient in the food and feed industries. The present work, for the first time, reported useful data on the effects of the SC-CO_2_ defatting process on lupin oil yield and quality. The results showed that, on average, the SC-CO_2_ defatting process prevented the total oil recovery from dipping below 60%, with significantly higher percentages for the two white lupin genotypes, namely Luxor and the Modica ecotype, in comparison to Multitalia and the other yellow and narrow-leafed genotypes. Based on the obtained results, the SC-CO_2_ technology can be proposed as a useful and chemical-free process for defatting lupin seed flour, allowing the acquisition of solvent-free lupin protein isolates and lupin oils to be conveniently used in the food and feed industries. The oil yield differed according to the species and genotype.

The total saturated fatty acid quantities were very similar between the samples. The most abundant saturated fatty acids were palmitic, stearic, and beric. The predominant monounsaturated fatty acid was oleic acid. Among the main polyunsaturated fatty acids, the most abundant were linoleic and alpha linoleic acid.

The total tocopherol quantities averaged about 1500.00 µg/g FW. This total amount of tocopherols was not sufficient to define their quality; nevertheless, the benefits of alpha-tocopherols should be considered, as they provide greater amounts of vitamin E, as well as gamma-tocopherols, which have high antioxidant activity in food. It can be concluded that lupin, in addition to being a good supply of mineral elements, also contributes vitamin E and powerful antioxidant activity, thanks to the very high content of gamma-tocopherols.

A detailed proteomic analysis of *Lupinus albus* was performed. The 2D gel analysis showed substantial equivalence between the cultivars of the same species (*Lupinus albus* and *L. luteus*). Moreover, the *L. angustifolius* proteome map showed the presence of additional spots in comparison to *L. albus,* corresponding to α-conglutins.

## Figures and Tables

**Figure 1 molecules-27-08771-f001:**
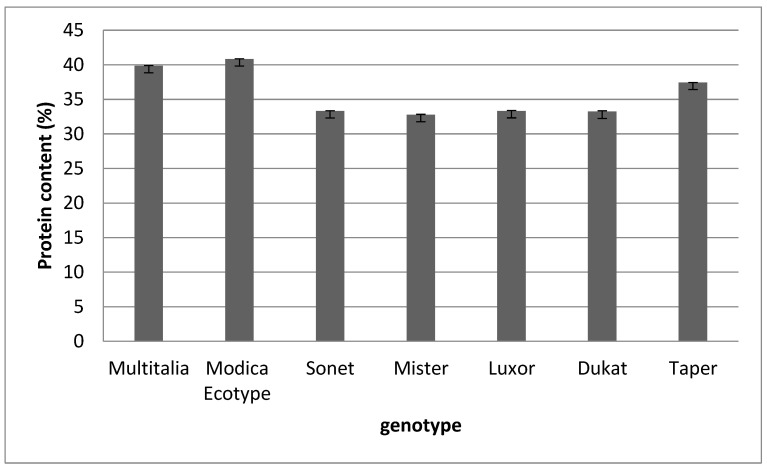
Protein content of the different *Lupinus* spp. genotypes. Data are presented as means ± standard deviation.

**Figure 2 molecules-27-08771-f002:**
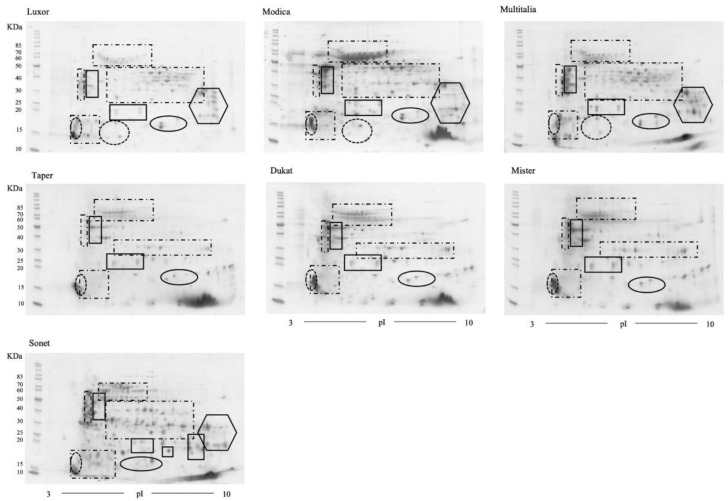
2D gels of three different cultivars of *Lupinus. albus* (upper gels), *L. luteus* (middle gels), and *L. angustifolius* (lower gel). Standard markers are indicated (in kDa) on the left side. Solid-line rectangles indicate the acidic and basic subunits of α-conglutin/legumin. Dash-and-dot rectangles indicate β-conglutin/vicilin. The dotted ovals indicate δ-conglutin. The solid-line ovals enclose the large and small subunits of γ-conglutin. The solid-line hexagons enclose the subunits of both β-conglutin and γ-conglutin.

**Figure 3 molecules-27-08771-f003:**
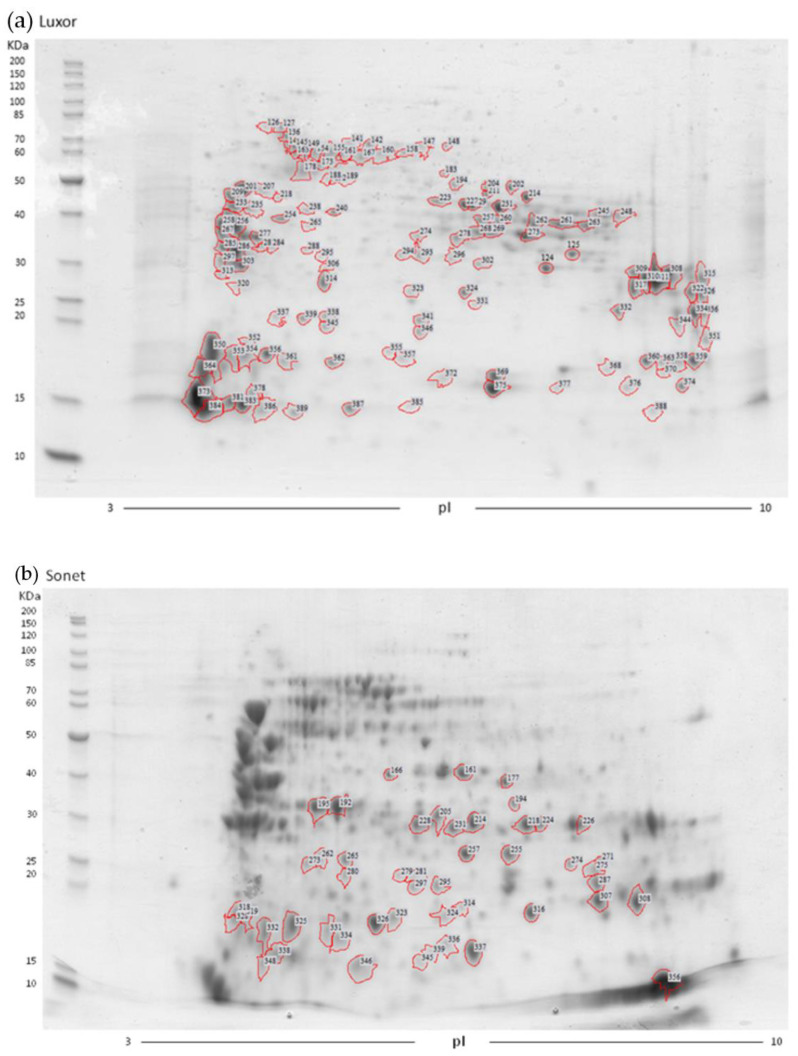
Excised protein spots are marked with red lines in (**a**) *L. albus* cv. Luxor and (**b**) *L. angustifolius* cv. Sonet.

**Figure 4 molecules-27-08771-f004:**
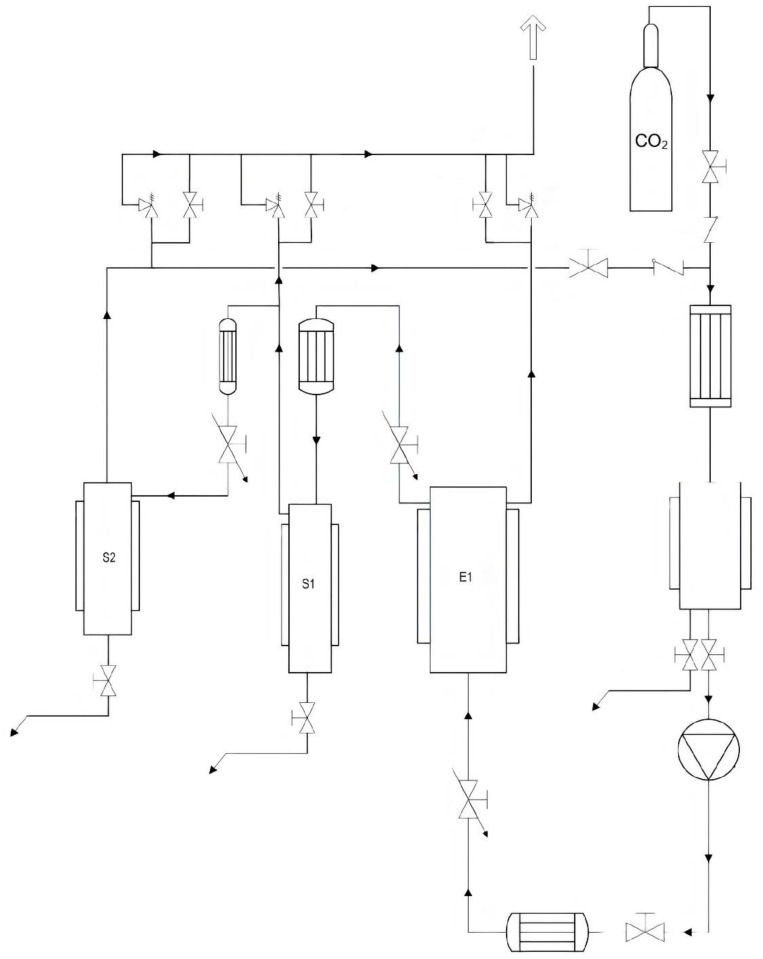
Scheme of the supercritical CO_2_ extraction plant used for defatting lupin seed flour samples.

**Table 1 molecules-27-08771-t001:** Content of micronutrients in different *Lupinus* spp. genotypes.

Species	Genotype	Micronutrients (mg/kg)			
		Cu	Fe	Mn	Zn
*Lupinus albus*	Luxor	10 ± 0.00 c	40 ± 0.00 d	185 ± 7.07 b	40 ± 0.00 b
	Multitalia	85 ± 7.07 a	55 ± 7.07 cd	235 ± 7.07 a	45 ± 7.07 b
	Modica ecotype	10 ± 0.00 c	55 ± 7.07 cd	110 ± 14.14 c	55 ± 7.07 ab
*Lupinus luteus*	Dukat	17 ± 1.41 c	200 ± 0.00 a	70 ± 0.00 d	75 ± 7.07 ab
	Mister	6.5 ± 0.70 c	125 ± 7.07 b	70 ± 0.00 d	70 ± 0.00 ab
	Taper	70 ± 0.00 b	160 ± 14.14 b	30 ± 0.00 e	90 ± 14.14 a
*Lupinus angustifolius*	Sonet	10 ± 0.00 c	80 ± 0.00 c	60 ± 0.00 de	45 ± 7.07 b

Data are expressed as means of three analytical replicates ± standard deviation; *p* ≤ 0.01—letters within the same column. Nutrient levels are herein expressed as mg kg^−1^ FW.

**Table 2 molecules-27-08771-t002:** Content of macronutrients in different *Lupinus* spp. genotypes.

Species	Genotype	Macronutrients (mg/kg)				
		K	P	Ca	Mg	Na
*Lupinus albus*	Luxor	1265 ± 63.63 d	5465 ± 106.06 b	2240 ± 42.42 bc	2200 ± 28.28 bc	435 ± 7.07 b
	Multitalia	690 ± 14.14 e	5385 ± 275.77 b	2350 ± 155.56 b	1990 ± 42.42 cd	360 ± 14.14 b
	Modica ecotype	1750 ± 28.28 c	5590 ± 339.41 ab	1620 ± 155.56 c	2300 ± 28.28 bc	375 ± 7.07 b
*Lupinus luteus*	Dukat	680 ± 14.14 e	6805 ± 35.35 a	2535 ± 7.07 b	1815 ± 91.92 d	360± 0.00 b
	Mister	1640 ± 14.14 c	6275 ± 120.20 ab	2555 ± 106.06 b	2825 ± 7.07 a	310 ± 14.14 b
	Taper	9015 ± 120.20 a	6825 ± 318.19 a	3200 ± 84.85 a	2315 ± 91.92 b	945 ± 63.63 a
*Lupinus angustifolius*	Sonet	2260 ± 0.00 b	6075 ± 219.20 ab	2635 ± 162.63 ab	2475 ± 49.49 b	345 ± 7.07 b

Data are expressed as means of three analytical replicates ± standard deviation; *p* ≤ 0.01—letters within the same column. Nutrient levels are herein expressed as mg kg^−1^ FW.

**Table 3 molecules-27-08771-t003:** Total fat content and experimental results of the different *Lupinus* spp. genotypes subjected to the SC-CO_2_ defatting process.

			SC-CO_2_ Defatting Process	
			(P = 300 bar; T = 40 ± 1 °C)	
		Fat	Oil	Oil
Species	Genotype	Content	Yield	Recovery
		(g/100 g DW)	(g/100 g DW)	(%)
*Lupinus albus*	Multitalia	18.08 ± 0.12 A	11.42 ± 0.31 A	63.15 ± 1.57 B
	Luxor	13.42 ± 0.66 B	9.66 ± 0.34 B	72.04 ± 1.10 A
	Modica ecotype	9.17 ± 0.18 D	6.55 ± 0.34 C	71.48 ± 3.82 A
*Lupinus luteus*	Dukat	10.46 ± 1.06 CD	6.49 ± 0.55 C	62.09 ± 1.25 B
	Mister	10.88 ± 0.75 BC	6.80 ± 0.43 C	62.54 ± 0.66 B
	Taper	7.88 ± 0.69 D	4.78 ± 0.40 D	60.70 ± 0.55 B
*Lupinus angustifolius*	Sonet	11.05 ± 0.50 BC	6.78 ± 0.24 C	61.35 ± 0.61 B

Data are presented as means ± standard deviation. Means in the same row followed by different letters are significantly different: *p* ≤ 0.01; n = 3 processing runs.

**Table 4 molecules-27-08771-t004:** Saturated fatty acid profiles of seed oil extracted from different *Lupinus* spp. genotypes.

Species	Genotype	C10:0	C12:0	C14:0	C16:0	C18:0	C20:0	C22:0	C24:0
*Lupinus albus*	*Luxor*	0.02 ± 0.01	0.03 ± 0.02	0.15 ± 0.05 ab	7.07 ± 0.49 ab	1.68 ± 0.10 c	1.20 ± 0.32 b	3.46 ± 0.47	0.94 ± 0.06
	*Multitalia*	0.02 ± 0.00	0.03 ± 0.00	0.12 ± 0.04 b	8.22 ± 0.34 ab	2.33 ± 0.30 bc	1.14 ± 0.03 b	4.04 ± 0.26	1.15 ± 0.17
	*Modica ecotype*	0.01 ± 0.01	0.02 ± 0.01	0.16 ± 0.03 ab	7.98 ± 0.35 ab	1.67 ± 0.14 c	1.13 ± 0.26 b	3.56 ± 0.33	1.22 ± 0.33
*Lupinus luteus*	*Dukat*	0.5 ± 0.03	0.02 ± 0.01	0.21 ± 0.03 ab	4.69 ± 0.82 b	3.84 ± 0.52 abc	3.79 ± 0.60 a	7.08 ± 0.32	1.13 ± 0.27
	*Mister*	0.09 ± 0.02	0.03 ± 0.05	0.23 ± 0.01 ab	9.54 ± 3.54 ab	5.05 ± 1.20 ab	1.75 ± 0.92 ab	3.51 ± 1.58	0.63 ± 0.12
	*Taper*	0.10 ± 0.08	0.04 ± 0.00	0.22 ± 0.00 ab	6.67 ± 0.70 ab	2.25 ± 0.67 bc	1.69 ± 0.74 ab	4.62 ± 0.64	1.12 ± 0.24
*Lupinus angustifolius*	*Sonet*	0.06 ± 0.05	0.03 ± 0.01	0.24 ± 0.00 a	11.05 ± 1.59 a	6.04 ± 0.29 a	1.56 ± 0.85 ab	3.39 ± 1.87	0.61 ± 0.21

Data are expressed as means of three analytical replicates ± standard deviation. Saturated fatty acid levels are herein expressed as g/100 g FW. Different letters in the same column indicate significant differences at *p* ≤ 0.01 or *p* ≤ 0.05.

**Table 5 molecules-27-08771-t005:** Monounsaturated fatty acid profiles of seed oil extracted from different *Lupinus* spp. genotypes.

Species	Genotype	C16:1n9	C18:1n9	C18:1n11	C20:1n11	C22:1n13
*Lupinus albus*	*Luxor*	0.42 ± 0.05 ab	49.41 ± 1.27 a	2.52± 0.05 a	4.05 ± 0.25 a	1.34 ± 0.20 ab
	*Multitalia*	0.48 ± 0.06 a	47.54 ± 2.63 ab	2.50 ± 0.02 a	4.30 ± 0.37 a	1.94 ± 0.20 a
	*Modica ecotype*	0.41 ± 0.02 ab	46.19 ± 3.02 abc	2.31± 0.04 a	3.56 ± 0.46 ab	1.53 ± 0.19 ab
*Lupinus luteus*	*Dukat*	0.14 ± 0.06 bc	26.47 ± 1.23 c	0.68 ± 0.02 b	1.52 ± 0.14 ab	0.84 ± 0.20 ab
	*Mister*	0.07 ± 0.02 c	27.80 ± 0.89 bc	0.66 ± 0.12 b	0.81 ± 0.56 b	0.25 ± 0.25 b
	*Taper*	0.26 ± 0.10 abc	34.89 ± 8.52 abc	1.64 ± 0.58 ab	2.96 ± 1.27 ab	1.23 ± 0.57 ab
*Lupinus angustifolius*	*Sonet*	0.06 ± 0.01 c	27.96 ± 0.45 bc	0.57 ± 0.01 b	0.59 ± 0.32 b	0.20 ± 0.19 b

Data are expressed as means of three analytical replicates ± standard deviation. Monounsaturated fatty acid levels are herein expressed as g/100 g FW. Different letters in the same column indicate significant differences at *p* ≤ 0.01 or *p* ≤ 0.05.

**Table 6 molecules-27-08771-t006:** Polyunsaturated fatty acid profiles of seed oil extracted from different *Lupinus* spp. genotypes.

Species	Genotype	C18:2n9.12	C18:3n9.12.15	C20:2n11,14	C20:3n3	C22:2n6
*Lupinus albus*	*Luxor*	18.10 ± 0.76	9.10 ± 0.18 c	0.31 ± 0.07 abc	0.18 ± 0.02 ab	0.00 ± 0.00 b
	*Multitalia*	17.48 ± 0.31	7.77 ± 0.17 c	0.35 ± 0.05 ab	0.24 ± 0.03 a	0.09 ± 0.03 ab
	*Modica ecotype*	19.65 ± 1.42	9.95 ± 0.36 bc	0.40 ± 0.04 a	0.22 ± 0.03 ab	0.09 ± 0.03 ab
*Lupinus luteus*	*Dukat*	38.94 ± 2.40	6.61 ± 0.25 ab	0.27 ± 0.09 abc	0.00± 0.00 b	0.15 ± 0.02 a
	*Mister*	39.99 ± 0.56	5.68 ± 0.56 a	0.11 ± 0.09 bc	0.06 ± 0.09 ab	0.06 ± 0.06 ab
	*Taper*	28.65 ± 8.97	9.26 ± 0.34 abc	0.32 ± 0.09 abc	0.21 ± 0.01 ab	0.13 ± 0.02 a
*Lupinus angustifolius*	*Sonet*	38.18 ± 2.31	5.65 ± 0.48 ab	0.08 ± 0.04 c	0.04 ± 0.05 ab	0.03 ± 0.04 ab

Data are expressed as means of three analytical replicates ± standard deviation. Polyunsaturated fatty acid levels are herein expressed as g/100 g FW. Different letters in the same column indicate significant differences at *p* ≤ 0.01 or *p* ≤ 0.05.

**Table 7 molecules-27-08771-t007:** Proximate tocopherol compositions of seed oil extracted from different *Lupinus* spp. genotypes.

Species	Genotype	Alpha-Tocopherol	Gamma-Tocopherol	Delta-Tocopherol	Total Tocopherols
	*Commercial flour of white lupin*	14.90 ± 2.12 b	1083.00 ± 18.38 ab	101.10 ± 1.77 ab	1300.09 ± 19.81 abc
*Lupinus albus*	*Luxor*	12.32 ± 1.60 b	1682.63 ± 33.41 a	109.62 ± 6.19 a	1914.19 ± 23.04 ab
	*Multitalia*	8.54 ± 1.04 b	841.86 ± 83.24 ab	49.36 ± 3.61 bcd	922.10 ± 91.50 cd
	*Modica ecotype*	12.57 ± 1.60 b	1608.47 ± 31.78 a	106.54 ± 6.42 ab	1834.12 ± 43.03 ab
*Lupinus luteus*	*Dukat*	8.59 ± 2.55 b	1377.20 ± 76.08 ab	14.95 ± 6.29 d	1415.70 ± 91.22 abc
	*Mister*	50.01 ± 28.93 b	1581.13 ± 209.68 a	71.06 ± 2.24 abcd	1773.26 ± 185.24 ab
	*Taper*	170.23 ± 10.10 a	1318.53 ± 417.52 b	74.47 ± 18.34 abc	1637.71 ± 464.30 abc
*Lupinus angustifolius*	*Sonet*	15.37 ± 7.45 b	583.45 ± 40.64 b	37.80 ± 1.47 cd	674.42 ± 30.244 c

Data are expressed as means of three analytical replicates ± standard deviation. Tocopherol levels are herein expressed as µg/g FW. Different letters in the same column indicate significant differences at *p* ≤ 0.01 or *p* ≤ 0.05.

**Table 8 molecules-27-08771-t008:** List of conglutins identified in the Luxor cultivar (*Lupinus albus*). Spots were grouped according to the identified amino acid sequence region.

Spot ID	Protein Identification	Acc. No.	Sequence Region
223, 257/260, 363/370	Vicilin-like protein	Q53HY0	N-terminal and central
204/211, 308	β-conglutin precursor	Q6EBC1	N-terminal and central
125, 136, 209, 231, 233, 238/265, 240, 245, 248, 262, 263, 268/269, 273, 274, 278, 286, 288/295/306, 294/293, 296, 326, 332, 350, 351, 373/384, 381, 383, 386, 389	Vicilin-like protein	Q53HY0	Central
124, 142/167/160, 145, 148, 149/163, 201, 202, 261, 284, 314, 315, 322, 336, 337, 344, 352/354, 356, 361	β-conglutin precursor	Q6EBC1	Central
313	β 2 conglutin	F5B8W0	Central and C-terminal
154, 334	β-conglutin	Q6EBC1	Central and C-terminal
155, 188/189/192, 207/218, 353, 267/285/258, 297, 303, 310, 317, 320, 158/147	Vicilin-like protein	Q53HY0	Central and C-terminal
302	β 2 conglutin	F5B8W0	
183/194	Vicilin-like protein	Q6EBC1	
214	β-conglutin precursor	Q6EBC1	
324, 374	BLAD (fragment)	Q0R0N3	Central
141/161, 178/173, 309, 311, 323, 364, 376	Vicilin-like protein	Q53HY0	
233, 235/254, 256	Legumin-like protein	Q53I54	N-terminal and central
341, 346	Legumin-like protein	Q53I54	Central
227, 359	α 2 conglutin	F5B8V7	C-terminal
368	Seed storage protein	F5B8V8	C-terminal
277	α 1 conglutin	F5B8V6	
202	α 2 conglutin	F5B8V7	
338, 345	Legumin-like protein	Q53I55	
308, 315, 322, 326, 334, 344, 357, 360, 363/370, 389	γ-conglutin	Q9FSH9	Central
296, 309, 310, 317, 376, 372, 377	γ-conglutin	Q9FSH9	Central and C-terminal
369, 374	γ-conglutin	Q9FSH9	C-terminal
204/211, 262, 257/260, 294/293, 324, 358, 368, 375, 388, 261	γ-conglutin	Q9FSH9	
378	γ-conglutin	Q9FEX1	Central and C-terminal
296, 350, 355, 364, 373/384, 381, 386, 387, 389, 385	δ-conglutin	Q333K7	Central and C-terminal
341, 388	δ-conglutin	Q333K7	C-terminal

**Table 9 molecules-27-08771-t009:** List of identified conglutins in Sonet cultivar (*L. angustifolius*). Spots were grouped according to the identified sequence region.

Spot ID	Protein Identification	Acc. No.	Sequence Region
323	β-conglutin	B0YJF7	N-terminal and central
308	β-conglutin	B0YJF7	Central
166, 177	β-conglutin	B0YJF8	Central
161, 271	β 5 conglutin	F5B8W3	Central
205, 228, 255, 275, 287, 346	β 2 conglutin	F5B8W0	Central
279/281/297/295	β 3 conglutin	F5B8W1	Central
194	β 1 conglutin	F5B8V9	Central
192	β-conglutin	B8Q5G0	Central
273	β 4 conglutin	F5B8W2	Central
195, 214, 334, 338/348	β 2 conglutin	F5B8W0	Central and C-terminal
231	β 3 conglutin	F5B8W1	Central and C-terminal
218, 224, 226, 257, 262, 265, 318/319/328, 280	β 4 conglutin	F5B8W2	Central and C-terminal
274	β 2 conglutin	F5B8W0	
325, 326	β 1 conglutin	F5B8V9	
332	β-conglutin	B0YJF7	
274, 275, 287, 279/281/297/295	α 1 conglutin	F5B8V6	Central and C-terminal
271	α 1 conglutin	F5B8V6	
307, 308	α 2 conglutin	F5B8V7	Central and C-terminal
314/324, 316	α 3 conglutin	F5B8V8	Central and C-terminal
336/339/345, 337, 346, 356	γ-conglutin	Q42369	Central and C-terminal

## Data Availability

The data presented in this study are available on request from the corresponding author.

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
