# Peer review of "Multielemental, Nutritional, and Proteomic Characterization of Different Lupinus spp. Genotypes: A Source of Nutrients for Dietary Use"

_molecules, 2022, doi:10.3390/molecules27248771_

Round 1

Reviewer 1 Report

The manuscript "Multielemental, nutritional and proteomic characterization of different Lupinus spp. genotypes: a source of nutrients for dietary use" by A. Spina et al. is devoted to the influence of genotypes of Lupinus ssp. (L. albus, L. angustifolius and L. luteus) on the composition of lupine seeds to assess the potential of lupine as a food crop. To this end, the seeds of 7 lupine genotypes were compared to determine the genotypic effects on the composition of micro- and macronutrients, proteins, fatty acids and tocopherols, as well as to characterize their total protein extracts by the proteomic analysis.

The manuscript is well and logically organized. At the same time, before accepting the manuscript, the authors must provide small clarifications:

1. The sentence should be improved (L. 100-101): Lupin seeds proteins are divided in albumins and globulins with the seconds, being 9-fold most abundant.

2. Start sentences with a capital letter (L. 110, 115, 125, 128): γ-Conglutin(s).

3. It is very difficult for the reader to perceive the results of the study due to the poor organization of the experimental material. Authors should group genotypes by Lupinus species in all Tables and Figures, as was done in the single most informative Table 4.

4. Tables are incorrectly numbered in the manuscript, Table 4 follows Table 2.

5. Unfortunately, the Supplementary Materials file was not available to the reviewer.

Author Response

The manuscript "Multielemental, nutritional and proteomic characterization of different Lupinus spp. genotypes: a source of nutrients for dietary use" by A. Spina et al. is devoted to the influence of genotypes of  Lupinus ssp. (L. albus, L. angustifolius and L. luteus) on the composition of lupine seeds to assess the 
potential of lupine as a food crop. To this end, the seeds of 7 lupine genotypes were compared to determine the genotypic effects on the composition of micro- and macronutrients, proteins, fatty acids and tocopherols, as well as to characterize their total protein extracts by the proteomic analysis.
The manuscript is well and logically organized. At the same time, before accepting the manuscript, the authors must provide small clarifications:
1. The sentence should be improved (L. 100-101): Lupin seeds proteins are divided in albumins and  globulins with the seconds, being 9-fold most abundant.
Introduction was implemented with additional descriptions on the main groups of proteins.
2. Start sentences with a capital letter (L. 110, 115, 125, 128): γ-Conglutin(s).
Capital letters were inserted.
3. It is very difficult for the reader to perceive the results of the study due to the poor organization of the  experimental material. Authors should group genotypes by Lupinus species in all Tables and Figures, as was done in the single most informative Table 4.
Answer: Thanks for your valuable comment. Tables have been formatted according to Table 4 (now table 3).
As suggested, the different genotypes have been grouped by Lupinus species.
4. Tables are incorrectly numbered in the manuscript, Table 4 follows Table 2.
Thanks very much. Due to a typo error, tables were incorrectly numbered after table 2. Table numbers are  now corrected.
5. Unfortunately, the Supplementary Materials file was not available to the reviewer.
Supplementary materials have been uploaded, we don't understand how this could happen. We apologize  for the inconvenience. We will try to reload them. 

Reviewer 2 Report

This paper dealt with multielemental, nutritional and proteomic characterization of different Lupinus spp. genotypes: a source of nutrients for dietary use. This paper will be of interest to the readers of the journal. This study contains valuable results and information. Some suggestions are as follows:

Abstract: Q: I suggest indicating the cultivars used in the abstract if possible and add a short conclusion of the study at the end of the abstract. 

Lines 579-580: Q: It appears that some incomplete table is shown there. Please revise it.   

Conclusion: Q: I suggest deleting lines 736-741 that are already mentioned in the introduction. 

Author Response

This paper dealt with multielemental, nutritional and proteomic characterization of different Lupinus spp.  genotypes: a source of nutrients for dietary use. This paper will be of interest to the readers of the journal. 
This study contains valuable results and information. Some suggestions are as follows:
Abstract: Q: I suggest indicating the cultivars used in the abstract if possible and add a short conclusion of  the study at the end of the abstract.
Done. The authors greatly appreciated the reviewer's suggestions 
Lines 579-580: Q: It appears that some incomplete table is shown there. Please revise it. 
Answer: The reviewer is right: the authors have eliminated the small table, which is superfluous and reported the information contained in it in the text. Therefore, they thank the reviewer for the improvement of the form.
Conclusion: Q: I suggest deleting lines 736-741 that are already mentioned in the introduction.
Done. The authors greatly appreciated the reviewer's suggestions

Reviewer 3 Report

This is a comprehensive and systematic study of the composition, proteomics, and micronutrients of Lupinus spp. These plants have high potential applications in the food industry. The manuscript is well written and the results are compared with the literature quite well. A few comments:

1. Please provide more experimental details on the proteomic analysis, particularly data analysis 

2. Please provide the reasons on the analysis has been done. For example, why this analysis is selected?

3. Please provide the reason for choosing SC-CO2 defatting process. what are the advantages?

Author Response

This is a comprehensive and systematic study of the composition, proteomics, and micronutrients of Lupinus spp. These plants have high potential applications in the food industry. The manuscript is well written and the results are compared with the literature quite well. A few comments:
1. Please provide more experimental details on the proteomic analysis, particularly data analysis
Thanks for the time spent reviewing the manuscript. More details regarding proteomic workflow are implemented in section 2.5.1. Most of the details have been previously reported in the Materials and  methods section. Moreover, additional information about data analysis is reported in Section 2.5.1
2. Please provide the reasons on the analysis has been done. For example, why this analysis is selected?
In the past lupins were mainly cultivated for their use as ruminant food and as green fertilizers but, nowadays, lupin production is increasingly being used for human nutrition. Thus, the interests of food and medical industries in lupin compounds have risen remarkably.
The considerable diversity of the Lupinus genus is thus still widely unexplored and unexploited regarding its potential for health and welfare. Indeed, physiological and health-promoting effects are highly influenced by 
protein structure. Therefore, the analysis of the diversity of lupin seed proteins, about their bioactive properties, will provide a deeper understanding of relationships between structure and bioactive properties aimed at improving lupin seed nutraceutical potential.
For the above reason, proteomic analysis performed with 2D gel approach was selected in addition to  multielemental, fatty acids and tocopherol analysis to provide a complete profile of different genotypes.
In the introduction section, some comments on this were already present “The knowledge of protein composition is helpful for understanding the relationship between the protein content and the nutritional and technological properties of foodstuff [37] and mass spectrometry coupled to electrophoresis and 
chromatographic separation is the method of choice for protein food analysis”. 
Thanks to the reviewer comments, some additional reasons related to proteomic analysis are reported and implemented with the above considerations.
3. Please provide the reason for choosing SC-CO2 defatting process. what are the advantages?
Thanks for the time spent reviewing the manuscript. SC-CO2 defatting process is herein reported as a sustainable, environmental friendly and feasible alternative to the use of chemical solvents to defat lupin flour to obtain lupin protein isolates. This was already expressed in the manuscript in the INTRODUCTION section (‘Taking into consideration that lupin seed is recognized as a protein-rich crop and that its use as protein source for dietary uses is of increasing interest, the employment of SC-CO2 extraction as defatting process can be 
proposed to address consumers’ requests of natural proteins isolated from natural resources through chemicals-free processes’) and in the CONCLUSION section (‘The employment of innovative green technologies which do not involve the use of organic solvents or other potentially toxic chemicals to defat lupin seed flour is a matter of increasing interest since consumers are more and more approaching to the consumption of natural proteins isolated from protein-rich crops. Moreover, lupin seed oil is a product of relevant interest due to its fatty acids and tocopherol profile which suggest its use as a valuable ingredient in the food and feed industries. The present work, for the first time, report useful data on the effects of the SCCO2 defatting process on the lupin oil yield and quality. ….. Based on the obtained results, the SC-CO2 technology can be proposed as a useful and chemicals-free process to defat lupin seed flour, allowing to obtain both solvent-free lupin protein isolates and lupin oils to be conveniently used in food and feed industries. The oil yield was different depending on the species and genotypes’).
However, the advantages of using SC-CO2 defatting process has been also added in the RESULTS section, by adding this sentence (‘The use of SC-CO2 to defat lupin seed flour have relevant advantages when the use of both defatted lupin protein isolates and lupin seed oil as food or feed ingredients is concerned. Indeed, this process allow to obtain lupin oil and lupin protein isolates free from residual chemical solvents, while avoiding the correlated pollution due to the massive use of chemical and toxic solvents’).
